# Dual-Responsive and Reusable Optical Sensors Based on 2,3-Diaminoquinoxalines for Acidity Measurements in Low-pH Aqueous Solutions

**DOI:** 10.3390/s23062978

**Published:** 2023-03-09

**Authors:** Elizaveta V. Ermakova, Anastasia V. Bol’shakova, Alla Bessmertnykh-Lemeune

**Affiliations:** 1Frumkin Institute of Physical Chemistry and Electrochemistry, Russian Academy of Sciences, Leninsky Pr. 31-4, Moscow 119071, Russia; 2Department of Chemistry, Lomonosov Moscow State University, 1-3, Leninskie Gory, Moscow 119991, Russia; 3Institut de Chimie Moléculaire de l’Université de Bourgogne, CNRS UMR 6302, Université Bourgogne Franche-Comté, 9 Avenue Alain Savary, BP 47870, CEDEX, 21078 Dijon, France; 4Laboratoire de Chimie, CNRS UMR 5182, ENS de Lyon, 46 Allée d’Italie, CEDEX, 69364 Lyon, France

**Keywords:** quinoxaline, pH sensor, colorimetry, fluorescence, paper test strips, Langmuir–Blodgett technique

## Abstract

This work is focused on the age-old challenge of developing optical sensors for acidity measurements in low-pH aqueous solutions (pH < 5). We prepared halochromic (3-aminopropyl)amino-substituted quinoxalines **QC1** and **QC8** possessing different hydrophilic–lipophilic balance (HLB) and investigated them as molecular components of pH sensors. Embedding the hydrophilic quinoxaline **QC1** into the agarose matrix by sol-gel process allows for fabrication of pH responsive polymers and paper test strips. The emissive films thus obtained can be used for a semi-quantitative dual-color visualization of pH in aqueous solution. Being exposed to acidic solutions with pH in the range of 1–5, they rapidly give different color changes when the analysis is performed in daylight or under irradiation at 365 nm. Compared with classical non-emissive pH indicators, these dual-responsive pH sensors allow for an increase in the accuracy of pH measurements, particularly in complex environmental samples. pH indicators for quantitative analysis can be prepared by the immobilization of amphiphilic quinoxaline **QC8** using Langmuir–Blodgett (LB) and Langmuir–Schäfer (LS) techniques. Compound **QC8** possessing two long alkyl chains (*n*-C_8_H_17_) forms stable Langmuir monolayers at the air–water interface, and these monolayers can be successfully transferred onto hydrophilic quartz and hydrophobic polyvinylchlorid (PVC) substrates using LB and LS techniques, respectively. The 30-layer films thus obtained are emissive, reveal excellent stability, and can be used as dual-responsive pH indicators for quantitative measurements in real-world samples with pH in the range of 1–3. The films can be regenerated by immersing them in basic aqueous solution (pH = 11) and can be reused at least five times.

## 1. Introduction

Luminescent materials attract living interest in both fundamental and applied science, holding great promise for the conversion of academic research into technological innovations [1,2,3]. Molecular materials are widely studied in this field because the structure of organic emitters can be adapted to target applications, among which sensorics are of particular interest. Luminescent properties of organic compounds are known to be highly sensitive to the environment of the luminophore. This limits the number of suitable solid supports and immobilization procedures appropriated for fabrication of emissive molecular materials. From a synthetic point of view, this restricted choice of fabrication methods often imposes laborious structural modifications of the molecular probe in order to prepare a suitable precursor of the desired emissive sensing material. Cutting-edge developments in sensing in aqueous media are focused on the development of multi-responsive sensors in which a luminescence response on the analyte binding is accompanied by signal transduction using another channel, e.g., variations of absorbance. Dual-responsive sensors allow for solving crucial problems in sensorics, such as increase in selectivity of the analyses, for instance.

Solid-state optical sensors based on organic molecules can be prepared by immobilization of chromogenic molecules into a polymer matrix or by grafting these molecules onto inorganic supports [4]. Another convenient synthetic approach to the sensory materials involves the preparation of Langmuir monolayers at the air–water interface followed by their assembling into Langmuir–Blodgett (LB) or Langmuir–Schäfer (LS) films [5,6,7,8]. This method allows for an aware control on the molecular orientation both in Langmuir monolayers and on solid surfaces. A consecutive deposition of monolayers on solid supports affords structurally organized films with a tunable thickness. When efficient transfer conditions are developed, the fabrication of sensors is easy and rapid and may be performed in an aqueous medium using a minimum amount of organic solvents and an expensive organic precursor. Additional benefits, such as an increase in selectivity, sensitivity, and shortening duration of analyses, can be achieved due to an appropriate supramolecular organization of the 2D systems (monolayers) and precise control on the number of monolayers deposited onto the solid surface [9].

Rapid and real-time monitoring of pH values is critical in clinical, industrial, and environmental situations, and most analyses should be performed in aqueous media [10,11,12,13]. Therefore, pH indicators suitable for a wide range of experimental conditions are required, which likely explains the constant research interest in the development of novel halochromic systems.

Quinoxaline derivatives absorbing the light in the visible region attract great attention as optical sensors. The ability of quinoxalines to change their optical properties when *N*-heterocycle is protonated was thoroughly investigated to obtain colorimetric and fluorescent pH chemosensors [14,15,16,17,18,19,20,21,22,23,24,25,26,27] because these compounds change the color in acidic media with pH < 4 in which many of pH indicators are insensitive [28,29,30,31,32]. Unfortunately, most of the quinoxaline derivatives reported elsewhere in the literature for pH monitoring can be used only in organic solvents or binary solvent mixtures and are inconvenient for real-world analyses. Water-soluble quinoxalines or solid-state devices (litmus strip, for instance) based on these compounds are required for practical usefulness. Surprisingly, halochromic materials based on quinoxalines are limited to chromogenic drop-cast films prepared by evaporation of chemosensor solutions on glass surfaces [18] and paper strips [20,22,24]. All these materials were used only for qualitative analyses of acidic vapors, with the exception of paper strips for a semi-quantitative determination of picric acid through fluorescence changes [20]. Despite Langmuir monolayers formed by quinoxaline derivatives [33] and cavitands [34], dendrimers [35] and polymers [36] bearing quinoxaline moieties were investigated, but they never attracted much attention in sensing likely due to quenching of their luminescence in Langmuir monolayers induced by strong π–π stacking of these heteroaromatic molecules [37].

Recently we reported that a water-soluble push–pull 2,3-bis[4-(methoxycarbonyl)phenyl]-6,7-bis[(3-aminopropyl)amino]quinoxaline (**QC1**, Figure 1) can be used as efficient dual-responsive optical (spectrophotometry and fluorescence) pH indicator in the pH range of 1–5 in real-life samples [38].

In this work, we investigated different strategies for preparation of pH-responsive materials based on 2,3-bis[4-(alkoxycarbonyl)phenyl]-6,7-bis[(3-aminopropyl)amine]quinoxalines to develop reusable sensors for visual semi-quantitative and quantitative instrumental pH analyses.

First, halochromic polymer films and paper test strips based on hydrophilic quinoxaline **QC1** were prepared by a sol-gel process involving agarose. The pH indicators thus obtained allow for dual-color visualization of pH in the range of 1–5. Next, to increase the accuracy of measurements, the amphiphilic halochromic compound **QC8** (Figure 1) was obtained and deposited onto solid supports using LB or LS techniques. Thin film sensors thus prepared can be used for visual dual-channel pH monitoring in the pH range of 1–3 and are also suitable for more accurate analyses using an appropriate instrumentation (spectrophotometer or fluorimeter). To our knowledge, this is the first example of preparation of emissive LB/LS films based on quinoxalines and their successful use as a sensor.

## 2. Materials and Methods

Synthesis of chemosensors **QC1** and **QC8** is reported in the Supporting Information.

### 2.1. Preparation of Polymer Films and Test Strips

A sol-gel process involving agarose was used for the preparation of polymer films and test strips from compound **QC1**. An aqueous solution (*c* = 0.2 mM) of quinoxaline **QC1** (1.1 mg) was added to a solution containing agarose, type I-B (0.15 g), and glycerol (0.5 g) in deionized water (10 mL). This solution (**Solution Q**) was heated to 90 °C and then stirred at this temperature for 3 min.

*Preparation of polymer films.* A hot (90 °C) **Solution Q** (2 mL) was poured into a Petri dish and a flat glass disc was accurately placed on its surface so that the liquid was sandwiched between the bottom of the Petri dish and this glass disc. These special conditions of cooling the reaction mixture to room temperature allow for a control on the film thickness. The glass disk was removed after 1 h, and polymer films with approximately 2 mm thickness were cut out and deposited in dimples of a Teflon plate.

*Preparation of test strips.* Cellulose filter paper with 8–12 μm pore size (“Belaya lenta”, Bashkhimservice, Ufa, Russia) was chosen for the preparation of pH-indicator paper test strips. Paper strips (10 × 40 mm) were immerged for 20 s into a hot **Solution Q** (8 mL, 90 °C). Then, the strips were dried in air for 10 min, placed in a microwave (700 W) for 30 s, and then dried in air for additional 8 h.

### 2.2. Preparation of Monolayers and LB/LS Films

Computer controlled 1000-2 KSV Minitrough (*l* × *w* = 36.4 × 7.5 cm) model (KSV Instrument Ltd., Helsinki, Finland) with a Teflon trough and polyacetal barriers equipped with a platinum Wilhelmy plate (Sensitivity ± 0.01 mN m^−1^) was used for the preparation of Langmuir monolayers. All measurements were carried out at room temperature (25 ± 1 °C). Deionized water (18.2 MΩ cm, pH~5.5) produced by a Vodoley cartridge purificator (SPE Himelektronika, Moscow, Russia) was used as a subphase and cleanliness of its surface was checked in blank compression experiments (surface pressure after compression process should not exceed 0.3 mN m^−1^). Monolayers were formed by spreading a freshly prepared 0.1 mM chloroform solution of **QC1** or **QC8**. The spreading solution was deposited in 2.5 µL portions using a Gilson “Distriman” micropipette (“Gilson S.A.S.”, Villiers-le-Bel, France). Delivered drops were uniformly distributed over a water surface in a checkerboard pattern. After spreading of 100 µL of this solution, chloroform was allowed to evaporate for 15 min, and the monolayer was compressed at a rate of 10 mm min^–1^.

Langmuir monolayers at the water surface with fixed pH were recorded under the same conditions using acidified (HCl) or basified (NaOH) water as a subphase. Each compression isotherm was recorded at least three times to ensure reproducibility of results.

LB films were deposited onto quartz slides, while the LS technique was used to prepare films on PVC and mica substrates. Transfers of monolayers were performed at the constant surface pressure of 20 mN m^−1^ by vertically immersing and withdrawing the solid substrate at the speed of 5 mm min^−1^ (LB films) or by horizontal deposition onto a solid substrate (LS films). The transfer ratio for the LB and LS films was no less than 0.95.

### 2.3. In Situ UV–Vis Absorption Spectroscopy of Langmuir Monolayers

*In situ* UV–vis absorption spectra of Langmuir monolayers at the air–water interface were recorded in the 200–750 nm wavelength range using an AvaSpec-2048 FT-SPU (Avantes, Apeldoorn, The Netherlands) fiber-optic spectrophotometer equipped with a 75 W DH-2000 deuterium–halogen light source and a CCD array detector. The UV–vis reflectometric probe with the fiber-optic diameter of 400 µm, combined with a 6-fiber irradiating bundle, was placed perpendicular to the studied surface at a distance of 2–3 mm. The signal reflected from the deionized water surface was used as a baseline.

### 2.4. UV–Vis Absorption and Fluorescence Measurements of Solutions and LB/LS Films

The UV–vis absorption spectra of solutions and LB/LS films deposited onto quartz and polyvinyl chloride (PVC) substrates were recorded on a Shimadzu-24 spectrometer (Kyoto, Japan) in a wavelength range of 200–900 nm. The fluorescence spectra of solutions and LB/LS films deposited onto solid substrates were recorded on a Shimadzu RF-5301 spectrofluorometer (Kyoto, Japan) in a wavelength range of 200–900 nm. Liquid samples were analyzed at 25 ± 1 °C in quartz cuvettes with an optical path length of 1 cm. All studied samples were irradiated at 420 nm.

Fluorescence quantum yields were measured at 25 °C by a relative method using fluorescein in ethanol (*Φ* = 92%) as a standard [39]. The following equation was used to determine the relative fluorescence quantum yield:*Φ*_x_ = *Φ*_st_ ((*F*_x_ · *A*_st_ · *η*_x_^2^)/(*F*_st_ · *A*_x_ · *η*_st_^2^)),(1)
where *A* is the absorbance (in the range of 0.01–0.1), *F* is the area under the emission curve, *η* is the refractive index of the solvents (at 25 °C) used for the measurements, and the subscripts x and st represent the studied and standard compounds, respectively. The following refractive index values are used: 1.362 for ethanol and 1.445 for chloroform. The quantum yields are given in Appendix A.

### 2.5. Atomic Force Microscopy (AFM) Investigations

AFM measurements were performed on MultiMode Nanoscope V (Veeco, Plainview, NY, USA) operating in tapping mode. Noncontact polysilicon cantilevers with high aspect ratio were used (TipsNano, Estonia, resonant frequency—120 kHz, Q-factor—350). Image processing was performed using Gwyddion (Chechen) software (Version 2.62).

### 2.6. Visual Determination of pH in Aqueous Solutions

First, pH of studied solutions was measured using portative Ecotest 2000 pH-meter with combined ESK-10601/7 glass electrode (Econix, Moscow, Russia). The electrode was calibrated with commercial buffers (pH = 4.01, 6.86, and 9.18) beforehand to determine the pH of the studied solutions. A series of standard solutions with pH 1.0, 2.0, 3.0, 4.0, and 5.0 was prepared by addition of aqueous HCl to deionized water. Next, an aqueous solution containing 0.1 mM of K^+^, Na^+^, Mg^2+^, Ba^2+^, Ca^2+^, Zn^2+^, Co^2+^, Cd^2+^, Pb^2+^, Ag^+^, Ni^2+^, Cu^2+^, Hg^2+^, and Al^3+^ perchlorates (pH 4.52) was prepared. Three other studied samples were freshly prepared lemon and lime juices (pH 2.45 and 1.95, respectively) and a kefir whey acidified by HCl to pH 1.20.

Then, the acidity of these samples was determined using dual-responsive optical sensors developed in this work. The pH-induced color changes of **QC1** in polymer films and test strips and of **QC8** in LB/LS films were evaluated visually at daylight (absorption channel) or using 365 nm LED (emission channel). For polymer films deposited in the dimples of a Teflon plate, aqueous solutions with different pH (0.5 mL) were added by a Gilson “Pipetman P1000L” pipette, and color of solid sensor was controlled during 30 s. Test strips and LB/LS films were immersed in aqueous solutions with different pH for 30 s. The color changes of all pH indicators were observed immediately, but the exposition of sensors to analyzed solutions was continued up to 30 s to gain insight into stability of materials under experimental conditions. Decolorization of films was never detected, and obtained pH values were in excellent agreement with those determined by the pH-meter.

## 3. Results

### 3.1. Synthesis of Aminoquinoxalines

Aminoquinoxalines **QC1** and **QC8** were chosen for the solid sensor preparation. While chromophore **QC1** is soluble in aqueous medium, **QC8** is an amphiphilic compound with structural parameters fitting well with LB and LS techniques, owing to the presence of two hydrophobic alkyl chains attached to a planar heteroaromatic core and an opposite arrangement of hydrophilic and hydrophobic residues in the molecule.

Aminoquinoxaline **QC1** was prepared according to the procedure previously reported by us (Figure 1) [38]. Its homologue **QC8** was synthesized using the reaction sequences shown in Figure 1 and Appendix A. First, 4,4′-bis(octyloxycarbonyl)benzil (**1b**) was prepared from methyl ester **1a** according a two-step procedure involving sequential hydrolysis and esterification reactions (Appendix A). Then, this compound was transformed in aminoquinoxaline **QC8** using a synthetic strategy developed by us for the preparation of **QC1,** as shown in Figure 1. The size of the carbon chain in the alkoxycarbonyl substituent did not influence on the courses of condensation and amination reactions, and compounds **QC1** and **QC8** were obtained in similar overall yields.

The newly synthesized compounds **1b**, **3b**, and **QC8** were unambiguously characterized by NMR, IR, UV–vis, and HRMS (ESI) analysis (Appendix A).

Electron absorption and emission spectra of **QC1** and **QC8** in chloroform were very similar, as shown in Appendix A and summarized in Appendix A. These compounds absorb and emit visible light, owing to the presence of a long and unsymmetrically substituted conjugated system in the molecules. Both compounds have proton-sensitive heteroaromatic nitrogen atoms (Figure 2) and are of interest for the development of pH-sensitive materials for measuring acidity of low-pH aqueous solutions.

Indeed, a stepwise addition of hydrochloric acid (up to pH 1) to aqueous solutions of **QC1** induces bathochromic shifts of absorption and emission bands which are visible by the naked eye in daylight or under irradiation by 365 nm LED (a standard UV source in the laboratories), respectively (Figure 2 and Appendix A) [38]. The protonation constants and protonation sequence were determined by us previously [38]. The protonation proceeds in two steps, as shown in Figure 2. The spontaneous protonation of primary amine groups remoted from the quinoxaline core in aqueous solutions renders **QC1** water-soluble below pH 8. Protonation of quinoxaline nitrogen atom is observed in the pH range of 1–5 and leads to large bathochromic shifts of both absorption and emission bands [38]. Thus, pH-induced color changes can be observed by the naked eye or monitored using appropriate instrumentation.

**QC8** is insoluble in aqueous media and its protonation was briefly investigated in water–methanol (1:1, *v*/*v*) solution. Since the structures of **QC1** and **QC8** differ only in the size of alkyl substituents, the changes of electron absorption and emission spectra of **QC1** and **QC8** with pH were very similar (Appendix A). On the basis of these data, we concluded that **QC8** is of interest to prepare layer pH-sensitive materials.

### 3.2. Polymer Films and Test Strips Based on Aminoquinoxaline **QC1**

Paper test strips are convenient for performing express analyses, in particular under “in field” conditions. Preparation of paper strips by dip coating is commonly used for immobilization of halochromic molecules. This is a rapid and economical method to fabricate pH indicators for performing qualitative or semi-quantitative analyses in aqueous media using hydrophobic dyes. However, water-soluble chromophores, such as aminoquinoxaline **QC1**, being immobilized by this approach are rapidly washed-off. In this regard, we investigated the preparation of polymer films and paper test strips based on aminoquinoxaline **QC1** by sol-gel technique using agarose as a gelling agent. The polymer film thus prepared was deposited on the filter paper and in dimples of a Teflon plate.

Solid sensors thus prepared were brightly colored and exhibited a high stability in aqueous media. A high washing resistance of these pH indicators can be likely explained by formation of numerous intermolecular hydrogen bonds between the polysaccharide matrix (agarose) and the polar **QC1** molecules which contain ten heteroatoms (N and O) suitable for hydrogen bonding with hydroxyl groups of the polysaccharide chains.

Both solid sensors changed their color immediately in the pH range of 1–5. The color changes observed in daylight and under illumination by 365 nm UV lamp in acidified deionized water (Figure 3 and Figure 4) were similar to those observed in these aqueous solutions employing water-soluble dye **QC1** (Figure 2 and Appendix A). We assume that they are caused by protonation of the quinoxaline ring.

The colored response of the sensor did not change when deionized water was replaced by a solution containing 14 metal perchlorates (K^+^, Na^+^, Mg^2+^, Ba^2+^, Ca^2+^, Zn^2+^, Co^2+^, Cd^2+^, Pb^2+^, Ag^+^, Ni^2+^, Cu^2+^, and Al^3+^, 0.1 mM of each salt; pH = 4.52), which indicates that the pH measurements were not disturbed by metal ion binding likely due to a low affinity of protonated diaminopropane residues to metal ions (Figure 3 and Figure 4).

Moreover, these solid sensors can be used for the analysis of real-life samples, such as lemon and lime juices (pH 2.45 and 1.95, respectively) and a kefir whey (acidified by hydrochloric acid to pH = 1.20) (Figure 3 and Figure 4). Both colors (in daylight and under illumination at 365 nm) observed for lemon juice and kefir whey were in good agreement with the expectations. In contrast, pH of lime juice can be determined only in daylight because the fluorescent response of both polymer films and test strips was significantly disturbed.

Thus, the dual-color visualization can be useful to increase accuracy and precision, diminishing the influence of an individual color sensibility and occasional organic and inorganic compounds which are commonly present in real-world samples and may disturb the optical responses of the chromophores.

It is also worth noting that use of a porous paper sheet as a solid support also leads to an increase of accuracy and precision of visual analyses, in particular in the pH range of 1–5 because the color changes of paper strips covered by the polymer film are easily seen by the human eye (Figure 3 and Figure 4).

The sensors can be rapidly regenerated by simple dipping them in basic solution with pH 11.0 and can be reused for pH measurements at least five times without any decrease in color intensity and loss of their efficiency (Appendix A).

However, these pH indicators are suitable only for semi-quantitative analysis since accurate spectrophotometric pH measurements cannot be performed using these flexible films having a ruffled surface.

### 3.3. Monolayers of Amphiphilic Aminoquinoxaline **QC8** at the Air–Water Interface and Their Transfer onto Solid Supports

To optimize the solid sensor fabrication and prepare thin films allowing for quantitative pH determination using spectrophotometric/fluorometric measurements, we investigated a deposition of aminoquinoxalines **QC1** and **QC8** onto solid surfaces through the formation of Langmuir monolayers at the air–water interface followed by their assembling into Langmuir–Blodgett (LB) or Langmuir–Schäfer (LS) films onto transparent solid supports. We have chosen this immobilization strategy because the thickness of LB and LS films can be precisely controlled by changing the number of transferred monolayers, which provides an opportunity to minimize the amount of chromophore required for obtaining stable instrumental and visible optical signals. Moreover, the transfer of the monolayers and optical properties of obtained films can be easily tuned, varying the nature of solid supports.

Our preliminary experiments demonstrated that compound **QC1** did not form Langmuir monolayers at the air–water interface because of its high solubility in water. Amphiphilic aminoquinoxaline **QC8** being deposited onto the surface of acidic and basic aqueous solutions (pH = 1.0–11.0) formed stable monolayers which could be compressed to the surface pressure of approximately 42 mN·m^−1^. Compression isotherms of **QC8** on the surfaces of aqueous solutions with pH 11.0, 5.5 and 1.0 are depicted in Figure 5a. The extrapolated molecular areas corresponding to linear sections of isotherms equaled 140 Å^2^/molecule for the monolayer formed at pH 1.0 and 130 Å^2^/molecule on water subphases with pH 11.0 and 5.5. This likely indicates a structural similarity of all these monolayers. According to computational calculations based on minimum energy equilibrium geometry estimated within Spartan package [40] (Figure 5a, inset), a molecule of aminoquinoxaline **QC8** should occupy approx. 225 Å^2^ being in the face-on orientation with respect to the surface and only approx. 45 Å^2^ when the perpendicular arrangement of molecules is observed. Thus, it appears that **QC8** molecules adopt an inclined orientation in the monolayer or quinoxaline fragments lie parallel to the surface, while two hydrophobic octoxycarbonyl substituents are directed away from the water surface. The absorption bands observed in *in situ* UV–vis absorption spectra (Appendix A) were broadened, compared with those observed in chloroform solution, but the absorbance maxima displayed approximately the same positions. As shown in Appendix A, absorbance intensity is increased when the surface pressure is gradually raised. This spectrophotometric behavior indicates that π–π stacking of heterocyclic fragments in the monolayers is rather weak because such intermolecular interactions can be observed spectrophotometrically as a shift of absorption bands. It appears that the presence in this molecule of two aryl substituents twisted with respect the quinoxaline core and two linear polyamine substituents attached to another aromatic ring significantly decreases π–π stacking of aromatic rings in the monolayers. The shape of compression isotherms and *in situ* UV–vis absorption spectra recorded at different compression pressures also reveal the absence of structural reorganizations of studied monolayers during compression cycles.

Surprisingly, maxima of absorbance bands of these three monolayers were pH independent and appeared at 290 and 430 nm over the entire of pH range in contrast to that which was observed in water–methanol (1:1, *v*/*v*) solution (Appendix A). Similar results were obtained when *in situ* UV–vis absorption spectra of **QC8** monolayer were recorded at the constant surface pressure of 10 mN m^−1^, with a gradual acidification of subphase from pH of 11.0 to pH of 1.0 (Figure 5b). In this experiment, two absorption bands were observed over the pH range of 11.0 to 3.0, and further acidification of the subphase led to the appearance of a small shoulder at 490 nm.

This low sensitivity of **QC8** monolayers to pH changes likely revealed its lower basicity in the monolayer, as compared with that of **QC1** and **QC8** in aqueous solutions. This is easy to understand considering a less efficient solvation of charged protonated species on the water surface.

These results demonstrated that the fabrication of sensing films using the aminoquinoxaline monolayers is challenging because transfer of the monolayers onto solid supports should be performed controlling the acid–basic properties of self-assembled systems thus obtained. Therefore, we investigated the film preparation by vertical deposition (LB method) of monolayers onto hydrophilic quartz and by their horizontal deposition (LS method) onto hydrophobic polyvinylchloride (PVC) supports.

The deposition onto a quartz substrate by LB method was performed at a constant surface pressure of 20 mN m^−1^ from the surface of three aqueous subphases with pH 11.0, 5.5, and 1.0. Light absorption properties of the films thus obtained were investigated visually (Appendix A) and spectrophotometrically (Appendix A). It was observed both by the naked eye and instrumentally that **QC8** monolayers formed on the subphase with pH = 1.0 cannot be transferred onto a quartz surface, likely due to electrostatic repulsion of triply positively charged species. A naked eye inspection of the LB film prepared at pH = 11.0 revealed that the film is inhomogeneous. The film of best quality was obtained using **QC8** monolayer formed at the surface of deionized water (pH = 5.5). This film was uniformly colored and displayed the highest absorbance intensity among three studied films. Thus, this aqueous subphase was used for further investigations and for preparation of multilayer LB and LS films.

To gain insight into the morphology of one-layer films of **QC8** by atomic force microscopy (AFM), **QC8** monolayers were also deposited by LB and LS methods onto a mica surface. Good quality AFM images were obtained only for the film prepared by the LS method (Figure 6). The images reveal that the films thus obtained are homogeneous at the mesoscopic scale. They are composed of elongated species with the predominant height of 4.5 and 9 nm. This film thickness is higher than that which is expected for a **QC8** monolayer (up to 2.5 nm for the film with perpendicular orientation of quinoxaline molecules; Figure 5a) and corresponds to the dimension of **QC8** bilayer (4.5 nm) and aggregates formed by two bilayers (9 nm). The formation of bilayer films on a mica surface is currently not understood, and we cannot exclude that this deviation of the film height from the value expected for a monolayer is caused by the presence of strongly bonded hydroxyl anions and water molecules in the aggregates because the film formed under these conditions likely is composed of doubly protonated **QC8** molecules (Figure 2). Noteworthy, amphiphilic porphyrins deposited onto a mica surface by LB method also formed films composed of individual nano-sized species, with a surface roughness that corresponded to a bilayer structure of the aggregates [41].

Such perforated films, with a tight surface coverage by halochromic molecules and easy accessibility of receptor to analytes, held promise for fabrication of photoactive multilayer sensing materials with a high surface area. Unfortunately, the film morphology is dependent on the nature of support, and films deposited onto practically important surfaces, such as quartz or PVC, for instance, cannot be analyzed by AFM technique.

Next, 30-layer films were successfully prepared applying surface pressure of 20 mN m^−1^ on both hydrophilic (quartz) and hydrophobic (PVC) substrates, according to LB and LS techniques, respectively. The reproducibility of this fabrication method was successfully checked by measuring absorption of the transferred films in three independent experiments. The films were homogeneously colored and emissive, and their exposure to aqueous solutions with pH lower than 3 for 30 s led to large bathochromic shifts of both absorption and emission bands in the visible region (Figure 7 and Appendix A). Despite the difference in the solid supports, sensing properties of two types of films thus obtained were very similar, as shown in Figure 7 and Appendix A. A linear dependence of absorbance and emission on pH value allows for quantitative pH measurements using both pH indicators (Figure 8 and Appendix A). Immersion of both films in aqueous solution with pH 4.52 containing 0.1 mM of various metal cations (K^+^, Na^+^, Mg^2+^, Ba^2+^, Ca^2+^, Zn^2+^, Co^2+^, Cd^2+^, Pb^2+^, Ag^+^, Ni^2+^, Hg^2+^, Cu^2+^, and Al^3+^) does not induce any changes of absorption and emission of light, as compared with spectra obtained in deionized water acidified up to this pH (Appendix A). pH values obtained for all real-life samples (an acidified kefir whey and lemon and lime juices) using these calibration curves were in a good agreement with values measured by standard pH electrode (Figure 8). It is important to note that the analysis of lime juice can be performed using both spectrophotometric and fluorescent responses, which was not possible using sensors based on agarose matrix. The environmental influence on the fluorescence response in the analysis of lime juice is likely less important in these sensors due to closed molecular packing in LB and LS films.

These spectral changes can also be observed by the naked eye in daylight and under UV lamp (*λ* = 365 nm) (Figure 9) allowing for easy and rapid semi-quantitative pH measurements. Both types of pH indicators exhibit excellent stability in the entire pH range and can be easily regenerated by immerging in a basic aqueous solution (pH = 11.0) and reused, as shown in Figure 10.

Comparing sensing properties of these pH indicators with those obtained by immobilization of quinoxaline **QC1** in agarose polymer matrix (Figure 3 and Figure 4), one can conclude that all of them are efficient for dual-color visualization of pH, but they have distinct differences. Sensors obtained by immobilization of **QC8** according to LB/LS methods allow one to perform quantitative pH monitoring in the pH range of 1–3, in which most known pH sensors are inapplicable. These techniques for the immobilization of quinoxalines provide a rational consummation of the chromogenic compound because the thin films containing only 30 layers can be used for reproductible pH measurements and can be regenerated more than six times.

In contrast, to perform visual semi-quantitative pH measurements, the immobilization of hydrophilic compound **QC1** in agarose matrix is the best strategy because pH indicators thus obtained rapidly give a visual response in a large pH interval (1–5).

As shown in Table 1 and Appendix A, development of pH-sensitive materials for acidic vapors and low-pH aqueous solutions has attracted considerable attention [18,20,22,24,27,42,43,44,45,46,47,48,49,50,51,52]. Many dyes were immobilized according to various strategies, but LB and LS techniques were not used in this field to our knowledge. Surprisingly, many of the solid-state sensors for pH measurements in the range of 1–6 reported elsewhere in the literature are suitable only for qualitative analysis (Appendix A). It is also noteworthy that pH indicators were commonly used in the analyses of gaseous samples. pH measurements in aqueous solutions are more challenging because they can be performed using only washing-resistant materials. The dual-responsive naked-eye indicators reported previously gave low accuracy and were used for qualitative analysis of acidic vapors [24,27,42,45,48,49]. Our solid sensors are more precise and also should be less dependent on the environmental factors, compared with the pH indicators working through the single fluorescence intensity changes.

## 4. Conclusions

In this work, we report the preparation of pH-responsive quinoxaline-based materials for acidity measurements in low-pH aqueous solutions (pH < 5). Our molecular design is based on the introduction of (1) electron donating (3-aminopropyl)amino residues at six and seven positions of the quinoxaline core, and (2) voluminous aryl substituents at two and three positions to maintain emission in materials. According to these design principles, a water-soluble quinoxaline **QC1** and amphiphilic compound **QC8** were prepared and investigated as molecular components of solid pH sensors.

Immobilization of the hydrophilic **QC1** into an agarose matrix allows us to prepare inexpensive polymer films and paper test strips for a semi-quantitative dual-color visualization of pH in aqueous samples with pH 1–5. Various (14 metal ions were tested) metal cations do not interfere with protons.

Next, focusing on quantitative pH measurements, we investigated the fabrication of nano-structured film materials based on quinoxalines. It was demonstrated that the introduction of long alkyl chains in the molecule is required for the fabrication of stable Langmuir monolayers at the air–water interface. The transfer of the **QC8** monolayers onto hydrophilic quartz and hydrophobic PVC was carried out using LB and LS techniques, respectively. The films thus obtained were emissive likely because the presence of voluminous aryl substituents prevents strong π–π stacking neighboring quinoxaline molecules. These films were successfully used as dual-responsive sensors for quantitative analysis of acidity in the pH range of 1–3 by instrumental methods (spectrophotometry and fluorescence). It is also worth noting that in terms of consumption of organic compounds, ease of handling and automation this immobilization technique is very seductive because the film composed of only 30 monolayers gives reproductible sensing results, can be easily stored, and can be regenerated more than six times.

## Data Availability

Not applicable.

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
