# Peer review of "Dual-Responsive and Reusable Optical Sensors Based on 2,3-Diaminoquinoxalines for Acidity Measurements in Low-pH Aqueous Solutions"

_sensors, 2023, doi:10.3390/s23062978_

Round 1

Reviewer 1 Report

In this work, Ermakova et al report the synthesis of a series of aminoquinoxalines and the development of reusable devices for pH detection based on anchoring dyes QC1 and QC8 onto different matrices depending on their hydrophilicity/lipophilicity. The final systems proved to be effective as chromogenic/fluorogenic sensors for acid pH detection (pH 1-5) under daylight and LED/365nm in aqueous media, including real samples, and in the presence of various interferents. The structural features of the QC1 and QC8 dyes and their consequences in the resulting sensors were interesting and well discussed. The reuse of sensors has been proven. The topic is interesting and fits the scope of the journal, and the experimental work was performed competently. I believe this work would be suitable for publication after minor revision, as described below:

- introduction and conclusion are too long and should be shortened

- chemical structures of the dyes mentioned in Table 1 can be added. In fact, this table should be replaced to the end of the work, and a comparison with the dyes reported in this work should be added

- experiments for QC1 and QC8 pKa determination should be added to the background material and discussed in the text

Author Response

We would like to thank the reviewer for high appreciation of our work and pertinent comments which are all useful for its improvement.

- introduction and conclusion are too long and should be shortened;

The introduction and conclusion have been shortened.

- chemical structures of the dyes mentioned in Table 1 can be added. In fact, this table should be replaced to the end of the work, and a comparison with the dyes reported in this work should be added;

Thank you for this comment. The chemical structures of the dyes mentioned in Table 1 and Table S2 were added. Table 1 was replaced and a brief discussion of the data was added in the text (page 13).

- experiments for QC1 and QC8 pKa determination should be added to the background material and discussed in the text

The protonation constants of QC1 were determined in our previous article which is cited by us (ref. 38, Ermakova, E.V.; Cheprakov, A.V.; Bessmertnykh-Lemeune, A. Chemosensors 2022, https://doi.org/10.1134/S2070205118010057). In this article, we also proposed the protonation sequence and use DFT calculations to prove it [38]. In this manuscript, these data are summarized in Figure 2.

The structures of aminoquinoxalines QC1 and QC8 differ only in the length of the alkyl substituent and these compounds should have similar pKa values. QC8 is insoluble in water and comparative protonation studies cannot be performed. To our opinion, these studies have low interest because pKa values in polymer films and LB films differ significantly from pKa in aqueous solutions. Thus, these additional comparative investigations which can be performed in water-methanol solvent mixtures, for example, were not carried out. However, we replaced Figure 2 and add a brief discussion in the text (page 6) to simplify reading.

Reviewer 2 Report

Reviewer: The author developed pH- responsive quinoxaline-based materials for acidity measurements in low pH aqueous solutions (pH < 5). The molecular component of pH sensors based on (3-aminopropyl)amino-substituted quinoxalines (QC1) and (QC8) possessing methyl and octyl ester has been designed and synthesized respectively. The hydrophilic quinoxalines (QC1) into an agarose matrix using sol-gel technique allows preparing cheap polymer films and paper test strips for a semi-quantitative dual-color visualization of pH in aqueous samples. The author has also investigated the fabrication of stable Langmuir monolayer using long alkyl chain qinoxaline QC8 materials. The LB and LS techniques have been used to carry out transfer of QC8 layers into hydrophilic quartz and hydrophobic PVC. These films have been successfully used as dual-responsive sensors for quantitative analysis of acidity by spectrophotometry and fluorescence methods. The important characteristics of this sensor has been the ease of handle, reproducibility-sensing results and can be easily stored. The paper is interesting and well written, the data are clear and the figures well organized. The strategy reported is new and interesting.

I recommend the publication of this paper after minor revision, as detailed in the following.

1. The author need to explain the protonation sequence of quinoxaline QC1   and corresponding color changes observed in DI water, real life samples and containing metal perchlorates from pH 1 to 5 as shown in figure 2 and 3.

2.  In page 9, Lines 312-313 "According to …..estimated with spartan package " requires a reference to support why they used this calculations to explain the orientation and arrangements of aminoquinoxaline QC8.

3. Using schematic representation, the author should explain the mechanism of quinoxalines molecules get trap into a solid support (polymer films and paper test strips) and not dissolve in aqueous medium during pH sensing since QC1 chromophore is highly soluble in aqueous medium.

4. The author should provide the mass spectra (HRMS) for newly reported molecules.

Author Response

We would like to thank the referee for appreciation of our experimental work and our writing. We are also grateful for all pertinent comments which are very useful to further improve our work.

  1. The author need to explain the protonation sequence of quinoxaline QC1 and corresponding color changes observed in DI water, real life samples and containing metal perchlorates from pH 1 to 5 as shown in figure 2 and 3.

We thank the referee for this important comment.

Protonation scheme and constants of QC1 in aqueous solutions was discussed in detail in our previous article which is cited in this manuscript (Ermakova, E.V.; Cheprakov, A.V.; Bessmertnykh-Lemeune, A. Chemosensors 2022, https://doi.org/10.1134/S2070205118010057). These data are summarized in Figure 2. Protonation of quinoxaline nitrogen atom is observed in pH range of 1.0 - 5.0. This protonation induces bathochromic shifts of bands in both absorption and emission spectra. The changes can be observed by naked eye in day light (yellow–red transition) and under illumination at 365 nm (green–orange transition), respectively. To simplify reading, Figure 2 was replaced and a brief discussion of the protonation sequence was added in the text (page 6).

The discussion of color changes observed in DI water, real life samples and aqueous solution containing metal ions was completed (pages 6 and 12).

  1. In page 9, Lines 312-313 "According to …..estimated with spartan package " requires a reference to support why they used this calculations to explain the orientation and arrangements of aminoquinoxaline QC8.

SPARTAN is a computation program which commonly used for molecular modeling. These quantum mechanical calculations (semi-empirical methods) are useful for visualization of large organic molecules. When Langmuir monolayers are investigated, these calculations help to conclude on the orientation of molecules in the monolayers since three dimensions of the amphiphilic organic molecules are generally very different. Despite a limited value of this approach, its usefulness was experimentally proved for many cases and these calculations are commonly used to get preliminary insight into the structure of QC8 monolayers at water-air interfaces and solid surfaces. The calculations can be replaced by molecular models but they are less precise in the conformational analysis.

The reference on the program was given in the text. We didn’t add references in which such calculations were used for the discussion of experimental data because any selection in the huge list of the works will be iniquitous (see for example, Jurak, M., Szafran, K., Cea, P., & Martín, S. (2021). Analysis of molecular interactions between components in phospholipid-immunosuppressant-antioxidant mixed Langmuir films. Langmuir, 37(18), 5601-5616; Ponce, C. P., Araghi, H. Y., Joshi, N. K., Steer, R. P., & Paige, M. F. (2015). Spectroscopic and Structural Studies of a Surface Active Porphyrin in Solution and in Langmuir–Blodgett Films. Langmuir, 31(50), 13590-13599.)

  1. Using schematic representation, the author should explain the mechanism of quinoxalines molecules get trap into a solid support (polymer films and paper test strips) and not dissolve in aqueous medium during pH sensing since QC1 chromophore is highly soluble in aqueous medium.

We thank the referee for this important comment. We revised the discussion and add the explanation (page 4). A high washing resistance of our polymer films can be explained by the formation of numerous intermolecular hydrogen bonds between polysaccharide matrix (agarose) and the polar QC1 molecules which contain ten heteroatoms (N and O). However, any schematic presentation of these interactions will be speculative because not all of heteroatoms of QC1 may be involved in hydrogen bonding. Thus, we prefer to give this description in the text form.

  1. The author should provide the mass spectra (HRMS) for newly reported molecules.

The data of HRMS (ESI) analysis have already been given in the text (Supporting information). We added all experimental HRMS (ESI) spectra as Figures (Figures S11-S15) in the revised Supporting information.